# Less is More: Improving LLM Alignment via Preference Data Selection

**Xun Deng**[14]**, Han Zhong**[2]***Rui Ai**[3]**, Fuli Feng**[5]*,**Zheng Wang**[4]**, Xiangnan He**[5]*
[1]University of Science and Technology of China, [2]Peking University,
[3]Massachusetts Institute of Technology, [4]Alibaba Cloud Computing
[5]MoE Key Lab of BIPC, University of Science and Technology of China
dx981228@mail.ustc.edu.cn, hanzhong@stu.pku.edu.cn, ruiai@mit.edu
{fulifeng93,xiangnanhe}@gmail.com, wz388779@alibaba-inc.com

## Abstract

Direct Preference Optimization (DPO) has emerged as a promising approach for aligning large language models with human preferences. While prior work mainly extends DPO from the aspect of the objective function, we instead improve DPO from the largely overlooked but critical aspect of data selection. Specifically, we address the issue of parameter shrinkage caused by noisy data by proposing a novel margin-maximization principle for dataset curation in DPO training. To further mitigate the noise in different reward models, we propose a Bayesian Aggregation approach that unifies multiple margin sources (external and implicit) into a single preference probability. Extensive experiments in diverse settings demonstrate the consistently high data efficiency of our approach. Remarkably, by using just 10% of the Ultrafeedback dataset, our approach achieves 3% to 8% improvements across various Llama, Mistral, and Qwen models on the AlpacaEval2 benchmark. Furthermore, our approach seamlessly extends to iterative DPO, yielding a roughly 3% improvement with 25% online data, revealing the high redundancy in this presumed high-quality data construction manner. These results highlight the potential of data selection strategies for advancing preference optimization.

## 1 Introduction

Reinforcement Learning from Human Feedback [RLHF; 6, 65] has emerged as a crucial technique for aligning Large Language Models (LLMs) with human preferences and values. Traditional RLHF implementations involve a two-stage process: reward model training based on preference data followed by reinforcement learning optimization. However, this approach presents significant computational challenges, requiring loading multiple model instances and extensive hyperparameter tuning.

As an alternative, [39] introduced Direct Preference Optimization (DPO), which streamlines the alignment process by directly optimizing the LLM policy from preference data. DPO has demonstrated comparable effectiveness while substantially reducing computational requirements compared to classical RLHF. Following DPO's introduction, numerous studies have proposed improvements through modified learning objectives [62, 1, 15] and iterative learning schemes [53]. While these **algorithmic** advances have shown promise, there remains a critical gap in our understanding of the **data-centric** aspects of preference learning: *what characteristics of preference data contribute most to model alignment?*

This work thoroughly studies the impact of preference data quality on DPO training, which is crucial for developing more efficient training strategies. In particular, we achieve both *improved performance*

---

*Corresponding author

39th Conference on Neural Information Processing Systems (NeurIPS 2025).

and *reduced computational costs* through strategic data selection. Our research makes three primary contributions:

(1) We prove in theory the necessity of data selection in the presence of exogenous noise. Specifically, the noise in the reward model may flip the preference between response pairs, leading to the emergence of the *parameter shrinkage* issue. Furthermore, we demonstrate that margin-based selection criteria can effectively address this issue by inducing *parameter inflation*.

(2) Driven by the theoretical results and the derived margin-maximization principle, we propose a Bayesian Aggregation for Preference data Selection (**BeeS**) strategy. **BeeS** incorporates signals from both external rewards and DPO implicit rewards, and deprioritizes a preference pair if it exhibits a low reward margin from any single reward source to mitigate potential noise. Through extensive experiments across diverse datasets and models, we show that this selection strategy shows two consistent advantages: it substantially reduces computational overhead via efficient data selection and improves model performance compared to training on the full dataset. In particular, on the UltraFeedback dataset and its variants, our method identifies a $10\%$ data subset for DPO training on LLama, Mistral, and Qwen series models, consistently achieving 3% to 8% point improvements on the AlpacaEval 2.0 benchmark relative to training on the complete dataset.

(3) Finally, we extend our data selection framework to iterative DPO settings, showing that selectively sampling online data can simultaneously lower computational costs and improve performance. In particular, we achieve 48.49% win rate and 54.99% length-control win rate on the AlpacaEval 2.0 benchmark using only $25\%$ of the online data for training.

Our findings provide both theoretical insights into the dynamics of preference learning and practical guidelines for more efficient DPO implementations. This work bridges an important gap between algorithmic innovation and data quality considerations in the context of LLM alignment.

## 1.1 Related Work

**Preference learning algorithms.** Reinforcement Learning from Human Feedback also known as dueling RL [37] or preference-based RL [5], has become a crucial component of recent Large Language Models (LLMs) such as ChatGPT [36]. While the classical RLHF pipeline traditionally uses Proximal Policy Optimization, several alternative approaches have been proposed. These include but not limited other RL-based training algorithms [28, 63], rejection sampling [11, 17], conditional supervised fine-tuning [31, 56, 60], and Direct Preference Optimization [39]. Among these alternatives, DPO has gained significant attention due to its simplicity and robust performance. Following the introduction of DPO, numerous works [62, 1, 15, 32, 44, 18, 54, 19, 51] have attempted to improve its performance by modifying the DPO objective.

**Data selection in LLM Fine-tuning.** Data selection is crucial in LLM post-training [48] for two key observations: post-training typically converges rapidly, and excessive data can degrade model performance through overfitting or exposure to toxic content [43, 10]. Recent research has focused on enhancing instruction tuning efficiency by identifying high-quality subsets from large instruction datasets [4, 25, 52], often adapting active learning query strategies [40] to assess sample uncertainty and diversity. However, data efficiency in preference learning remains relatively unexplored. Prior studies have studied reducing annotation costs in preference dataset creation [35, 57] and on scenarios involving numerous ranking annotations [45, 34]. Other research aims to improve a model's ability to distinguish between two responses by adding a margin to the loss term [32, 38, 1]. Additionally, concurrent work highlights the importance of margins for preference data filtering, though there is debate on whether hard samples help or hinder preference learning [50, 22, 59, 16].

Our work firstly provides clear criteria for identifying informative samples while filtering toxic ones, thereby improving both DPO's efficiency and performance. Furthermore, our method extends to iterative DPO [53] and its variants [61], wherein training data is dynamically generated by the model during its iterative training process.

## 2 Background

Reinforcement Learning from Human Feedback (RLHF) has emerged as a key method for aligning LLMs with human preferences. It leverages training data of the form $\mathcal{D} = \{x, y_w, y_l\}$, where $x$ rep-

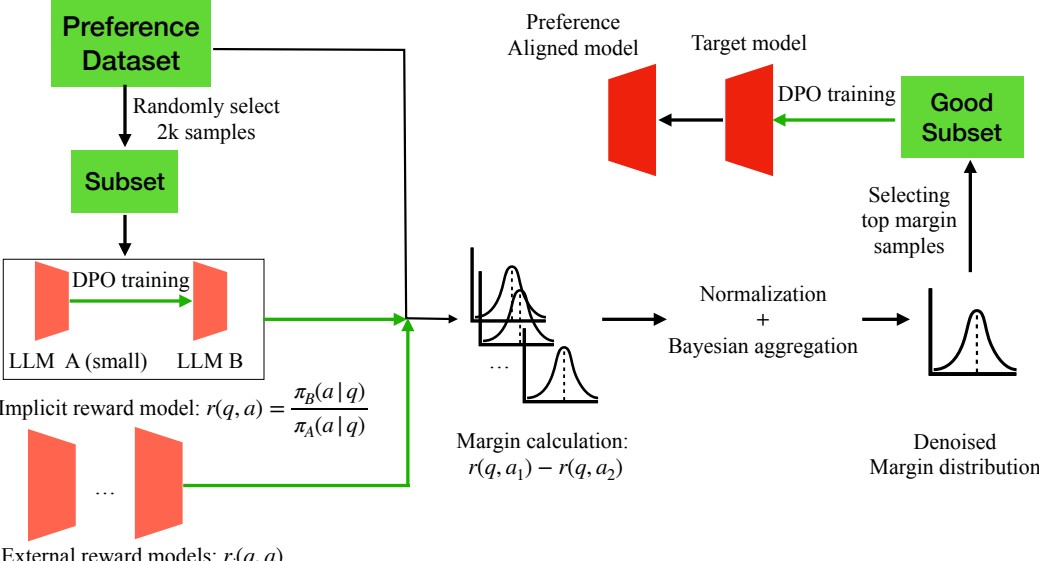

Figure 1: The workflow of the BeeS method.

resents the input prompt, and $y_w$ and $y_l$ denote the preferred and dispreferred responses, respectively. The RLHF pipeline typically involves two stages: reward learning and policy optimization.

**Reward Learning.** In the reward learning stage, a reward model is trained to approximate human preferences based on preference data. By adopting the Bradley-Terry model [3] to capture human preference, reward training involves minimizing the loss:

$$\mathcal{L}_{\text{RM}}(r) = -\mathbb{E}_{(x, y_w, y_l) \sim \mathcal{D}} \big[ \log \sigma \big( r(x, y_w) - r(x, y_l) \big) \big],$$

where $\sigma(\cdot)$ is the sigmoid function.

**Policy Optimization with Reinforcement Learning.** Once the reward model $r$ is trained, it is used to guide the optimization of a policy $\pi_\theta(y|x)$, where $\theta$ denotes the parameters of the model. This stage often employs reinforcement learning techniques such as Proximal Policy Optimization [PPO; 41] to optimize the policy by maximizing the expected reward.

$$\max_{\pi_\theta} \mathbb{E}_{x \sim \mathcal{D}, y \sim \pi_\theta(\cdot|x)} \left[ r(x, y) - \beta \log \frac{\pi_\theta(y|x)}{\pi_{\text{ref}}(y|x)} \right],$$

where $\beta > 0$ is the regularization parameter. However, this RL approach can be computationally expensive, sensitive to reward misspecification and require careful hyperparameter tuning.

Recently, as an alternative to the RL-based policy optimization in RLHF, *Direct Preference Optimization* [DPO; 39] has been proposed. DPO simplifies the reward alignment process by directly incorporating human preference data into supervised training. Instead of defining and optimizing a reward function explicitly, DPO minimizes

$$\mathcal{L}_{\text{DPO}}(\theta) = -\log \sigma \left( \beta \log \frac{\pi_\theta(y_w|x)}{\pi_{\text{ref}}(y_w|x)} - \beta \log \frac{\pi_\theta(y_l|x)}{\pi_{\text{ref}}(y_l|x)} \right).$$

By bypassing the intermediate step of reinforcement learning, DPO offers a more stable and computationally efficient alternative to standard RLHF, while still aligning models effectively with human feedback.

## 3 Methodology

### 3.1 Parameter Shrinkage and Inflation Analysis

We follow the model from [64] to illustrate why data selection can improve model performance. We assume that reward model $r(x, y) = \langle \phi(x, y), \omega^* \rangle$ with some feature function $\phi(\cdot, \cdot)$. For reward

Table 1: Symbols used in the formulation.

| **r** | **x** | $y_w/y_l$ | $\phi/\Delta\phi$ | $\zeta$ | **w** |
|---|---|---|---|---|---|
| Reward | Input prompt | Preferred/dispreferred Response | (relative) feature function | Exogenous error | Learnable parameters |

learning, our reward model can be an explicit $r(x, y)$ [36], while for DPO, $\beta \log \frac{\pi_\theta(y|x)}{\pi_{\text{ref}}(y|x)}$ plays the role of reward model implicitly [39]. Based on observations in previous literature, we can derive such features by removing the last layer of the pre-trained model. However, both humans and other large models may use inaccurate reward functions to generate labels, where the labels represent the ranking of two responses. We say the preference between $y_w$ and $y_l$ is generated by $r(x, y_w) - r(x, y_l) + \zeta$ where $\zeta$ is an exogenous error. We use $\Delta\phi(x)$ to denote $\phi(x, y_w) - \phi(x, y_l)$ for simplicity.

*Parameter Shrinkage.* Here, we hope to find $\omega$ to minimize

$$\mathcal{L}_{\text{RM}}(\omega) = -\mathbb{E}_{x,\zeta}\Big[\frac{1}{1 + e^{-\langle\Delta\phi(x),\omega^*\rangle-\zeta}}\log(\frac{1}{1 + e^{-\langle\Delta\phi(x),\omega\rangle}}) + \frac{1}{1 + e^{\langle\Delta\phi(x),\omega^*\rangle+\zeta}}\log(\frac{1}{1 + e^{\langle\Delta\phi(x),\omega\rangle}})\Big]. \tag{1}$$

It holds that the first-order condition is

$$\mathbb{E}_{x,\zeta}\Big[\frac{1}{1 + e^{\langle\Delta\phi(x),\omega^*\rangle+\zeta}}\frac{e^{\langle\Delta\phi(x),\omega\rangle}\Delta\phi(x)}{1 + e^{\langle\Delta\phi(x),\omega\rangle}}\Big] = \mathbb{E}_{x,\zeta}\Big[\frac{1}{1 + e^{-\langle\Delta\phi(x),\omega^*\rangle-\zeta}}\frac{e^{-\langle\Delta\phi(x),\omega\rangle}\Delta\phi(x)}{1 + e^{-\langle\Delta\phi(x),\omega\rangle}}\Big]. \tag{2}$$

Since we know that $\langle\Delta\phi(x), \omega^*\rangle$ is positive, when $\zeta$ is small comparing to the margin, it holds that $\frac{1}{1+e^{\langle\Delta\phi(x),\omega^*\rangle+\zeta}}$ is convex with respect to $\zeta$. Due to Jensen's inequality, it holds that

$$\mathbb{E}_{x,\zeta}\Big[\frac{1}{1 + e^{\langle\Delta\phi(x),\omega^*\rangle+\zeta}}\frac{e^{\langle\Delta\phi(x),\omega\rangle}\Delta\phi(x)}{1 + e^{\langle\Delta\phi(x),\omega\rangle}}\Big] \geq \mathbb{E}_x\Big[\frac{1}{1 + e^{\langle\Delta\phi(x),\omega^*\rangle}}\frac{e^{\langle\Delta\phi(x),\omega\rangle}\Delta\phi(x)}{1 + e^{\langle\Delta\phi(x),\omega\rangle}}\Big].$$

Similarly, we have

$$\mathbb{E}_{x,\zeta}\Big[\frac{1}{1 + e^{-\langle\Delta\phi(x),\omega^*\rangle-\zeta}}\frac{e^{-\langle\Delta\phi(x),\omega\rangle}\Delta\phi(x)}{1 + e^{-\langle\Delta\phi(x),\omega\rangle}}\Big] \leq \mathbb{E}_x\Big[\frac{1}{1 + e^{-\langle\Delta\phi(x),\omega^*\rangle}}\frac{e^{-\langle\Delta\phi(x),\omega\rangle}\Delta\phi(x)}{1 + e^{-\langle\Delta\phi(x),\omega\rangle}}\Big].$$

Since the optimal $\omega$ is $\omega^*$ without $\zeta$, plugging $\omega^*$ in Equation (2) will cause the left-hand side to be greater than the right-hand side. Therefore, the optimal $\omega$ with the existence of $\zeta$ intends to shrink to the original point compared to $\omega^*$ so that the first-order condition is still satisfied.

We provide the underlying intuition with an extreme example. If $\mathbb{V}(\zeta)$ goes to infinity, the preference between $y_w$ and $y_l$ mainly depends on $\zeta$, approaching a Rademacher distribution, then $\omega = 0$ could be a good solution to Equation (1). In other words, $\zeta$ offsets part of the information provided by the reward model, causing the model's parameters to shrink toward zero. Thus, data selection is essential for acquiring policies with good performance. Finally, we remark that $\zeta$ can come from multiple resources, including human classification errors, different embeddings or reward models from other LLMs and so on.

*Parameter Inflation.* We next explain why selecting data points based on the margin can lead to parameter inflation, thereby offsetting the parameter shrinkage caused by errors.

First, when the margin is large, namely, $\langle\Delta\phi(x), \omega^*\rangle + \zeta$ is large, from the S-shaped graph of $\sigma(\cdot)$, we know that the slope is very small in this area. As a result, the probability of preference reversal caused by $\zeta$ is low, which means the likelihood of incorrect samples is also low. Secondly, given prompt $x$, as we select data with large $\langle\Delta\phi(x), \omega^*\rangle + \zeta$, the posterior distribution of $\zeta$ is skewed toward the positive side. Therefore, the preferences corresponding to this kind of data are more pronounced, leading to inflated estimates of $\omega$ in Equation (1). Finally, we point out that if realized $y_w$ and $y_l$ are all separable, proportional scaling of $\omega$ can reduce the value of Equation (1) continuously. Hence, some techniques like regularization or early stopping when training are indispensable.

In summary, inaccuracies in the reward model can cause the parameters of LLMs to shrink toward zero. By selecting data with larger margins, we can compensate for the performance degradation caused by this shrinkage. The balance between parameter shrinkage and inflation offers the potential to enhance the performance of LLMs. Driven by this theoretical result, our main idea is to let the model reach an overall high margin during preference learning. We realize this by providing a robust estimation of reward margins for the entire dataset ahead of full-set training, which then allows efficient high-margin data filtering.

## 3.2 Multi-source Margin Aggregation

Building on our previous analysis, we aim to develop a data selection strategy based on the margin-maximization principle, with the calculation of reward margin being the critical component. We

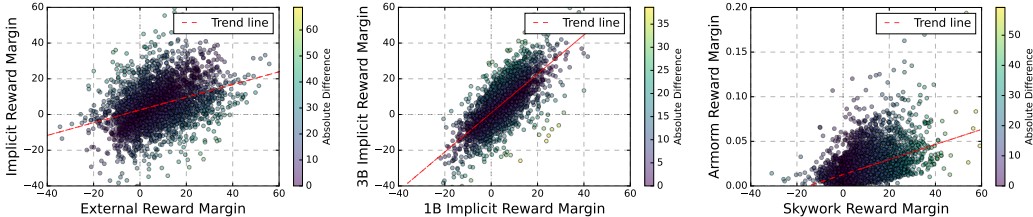

Figure 2: Visualization of joint margin distribution on **UltraFeedback**. (Left) Joint distribution of external and implicit reward margin values. (Middle) Joint distribution of implicit reward margins computed using models of 1B and 3B scales. (Right) Joint distribution of two different external reward margin values on online-generated data.

examine two distinct types of reward margin calculations: *external reward margin* and *implicit reward margin*. The external reward margin is provided by an external reward function, while the implicit reward margin is derived from the implicit reward $\log \frac{\pi_\theta}{\pi_{\text{ref}}}$, where $\pi_\theta$ represents the policy trained by DPO [39] (see Section 4.1 for details on margin calculation). We visualize the joint distribution of different reward margin sources using the UltraFeedback dataset [7] in Figure 2.

- The left and middle panels of Figure 2 reveal several key phenomena: (1) The correlation between implicit and external reward margins is notably weak. In particular, samples exhibiting high positive implicit margins span a broad range of external margins, including strongly negative values, and vice versa. This divergence highlights the distinct preference patterns captured by these two reward types. This underscores the need to combine both reward types for a reliable margin estimation. (2) In contrast, we observe a strong correlation between implicit reward margins calculated by models of different sizes (Llama-3.2 3B and 1B). Notably, these two patterns remain consistent across other datasets as well (see Appendix B.2).

- Online RLHF [53, 13] employs the target model to generate multiple responses for given prompts iteratively and uses an external reward model to identify the response pair with the largest margin for DPO training. The right panel of Figure 2 illustrates that a max-margin pair construction method, even when derived from one strong reward model, can still yield ambiguous preferences when evaluated by another reward model (which shows similar performance on RewardBench Leaderboard). This ambiguity in preference signal indicates that the online data generation process can still cause high redundancy, which may offer little to no benefit, or could even be detrimental, to online-DPO training.

These observations highlight the need for multi-source margin aggregation to achieve a more robust margin estimation, thereby enhancing data selection and preference learning. To this end, we propose a strict aggregation strategy, Bayesian Aggregation for Preference data Selection (**BeeS**), that deprioritizes a preference pair if it exhibits a low reward margin from any single reward source. We implement this method through a general three-step procedure.

*Step 1: in-distribution pre-DPO training.* Our objective is to obtain the in-distribution implicit reward model with low computational and GPU-memory cost. Given the strong correlation of that margin across different models (See Figure 2), this weak-to-strong guidance is feasible. To achieve this, we randomly select a small seed dataset $\mathcal{D}_0$ from $\mathcal{D}$ and employ a (or several) small model to perform preference learning on this seed set. The high sample and training efficiency of the DPO loss [23] ensures the feasibility of this approach.

*Step 2: margin calculation.* We calculate external and implicit reward margins as $m_{\text{ex}} = r_{\text{ex}}(x^i, y_w^i) - r_{\text{ex}}(x^i, y_l^i)$ and $m_{\text{im}} = r_{\text{im}}(x^i, y_w^i) - r_{\text{im}}(x^i, y_l^i)$ for each datum in $\mathcal{D}$, where we directly calculate $r_{\text{im}}(x, y)$ as $\log \frac{\pi_\theta(y|x)}{\pi_{\text{ref}}(y|x)}$. Here, $\pi_{\text{ref}}$ and $\pi_\theta$ denote the small model before and after preference learning. We assume that there are $K$ margins involved from these two typical sources.

*Step 3: Bayesian aggregation.* To mitigate noise from individual reward sources, we utilize Bayesian probability theory for their robust aggregation. Given that reward margins from different sources often vary in their underlying distributions and value ranges, we propose projecting these diverse margins into a unified probability space. This projection serves to quantify the confidence that a specific preference direction $y_w > y_l$ is correct, formally expressed as $\mathbb{P}(y_w > y_l | m^1, m^2, \cdots, m^K)$. Assuming that these sources are conditionally independent, then following the Bayesian formula

transformation in previous work [30, 9], the preference probability can be expressed as:

$$\mathbb{P}(y_w > y_l | m^1, m^2, \cdots, m^K) = \frac{\prod_{i=1}^{K} \mathbb{P}(y_w > y_l | m^i)}{\prod_{i=1}^{K} \mathbb{P}(y_w > y_l | m^i) + \prod_{i=1}^{K}(1 - \mathbb{P}(y_w > y_l | m^i))}. \quad (3)$$

The typical absence of well-defined preference labels (e.g., $y_w > y_l$ for clear preference, or $y_w = y_l$ for indifference) renders the rigorous estimation of single-source preference probabilities challenging.[2] Consequently, we approximate the probability using a linear projection: $\mathbb{P}(y_w > y_l | m^i) = \frac{\text{clip}(m^i, L, U) - L}{U - L}$ , where $\text{clip}(m, L, U) = \min(\max(m, L), U))$ and $L$, $U$ are tuning parameters. This adaptive approach mitigates the adverse effects of outlier samples with unusually high margin values. See more implementation details and discussion about the derivation approximation of Eq. (3) in Appendix A.1.

**Sample selection.** We consider the data selection for both the one-pass DPO training and the iterative DPO workflow. For the former, we directly select the samples with the highest aggregated preference probability to construct $\mathcal{D}_{\text{train}}$. The threshold depends on how many preference samples we prefer to use (but should guarantee that samples with negative margins are excluded). For the latter, we only need to train implicit reward models in the first iteration, and **BeeS** three-step procedures are applied for each iteration.

# 4 Experiments

We organize the experiments as follows: we explain the experimental setup in Section 4.1; we compare **BeeS** with various sample selection baselines on diverse preference tasks and present the detailed results in Section 4.2; then we focus on the important chat task, and explore the effectiveness of **BeeS** in enhancing comprehensive dialogue ability in Section 4.3. Lastly, we perform diverse ablation studies for the **BeeS** in Section 4.4.

## 4.1 Experimental Setup

**Preference Datasets.** We evaluate our approach using three established preference datasets: (1) Reddit **TL;DR** summarization dataset [47, 42] that contains human-written summaries and human-rated results, (2) Anthropic Helpful and Harmless dialogue dataset (**HH**) [2], and (3) **UltraFeedback** [7], which comprises quality-scored model responses across diverse prompts from multiple sources. To explore how models react to on-policy data, we leverage two modified versions of the **UltraFeedback** dataset, **Llama-UltraFeedback** and **Mistral-UltraFeedback** [32]. In the variants, the original chosen and rejected responses are replaced with the highest and lowest scored responses, respectively, sampled from five candidates generated by the corresponding Instruct model. The scores are given by the PairRM [21] reward model. Statistics about these datasets are in Appendix A.

**Models.** Our experiments are conducted across four model series: Llama-3.2 [33], Llama-3 [14], Mistral-7B-v2 [20], and Qwen-2.5 [55] under Base and Instruct setups. For the Base model (Llama-3.2-3B and Llama-3-8B), we first establish fundamental Instruction-following capabilities through supervised fine-tuning on the RLHFlow/SFT-OpenHermes-2.5-Standard datasets. For the Instruct setup, we directly use them as the start of DPO training. Regarding the external reward model, we adopt the recent Skywork-Reward-Llama-3.1-8B-v0.2 [29] that is the best reward model at this scale according to the RewardBench leadboard. As for the implicit reward model, we employ the Llama-3.2-3B Base and its DPO-tuned model (on 2,000 randomly selected samples from the complete dataset) for $\pi_{\text{ref}}$ and $\pi_{\theta}$.

**Implementation and Evaluation.** For DPO training, we follow [39] and use a fixed value of $\beta = 0.1$, except for **TL;DR** where $\beta = 0.5$. We run each training for two epochs, with a learning rate of $5 \times 10^{-7}$, and a 0.1 warmup ratio. Following [39], we evaluate the models using 400 randomly sampled test sets from the validation/test pools of the **TL;DR** and **HH** datasets, separately. For models trained on **UltraFeedback**, we employ AlpacaEval and AlpacaEval 2.0 [27] as our evaluation benchmark, which consists of 805 diverse questions.[3] As the ground truth oracle is unavailable,

---

[2]This estimation typically relies on methods such as isotonic regression or histogram analysis.

[3]They use the same set of questions, and differ in their reference response generation: AlpacaEval uses Text-Davinci-003 [58], whereas AlpacaEval 2.0 employs GPT4-1106-preview

Table 2: GPT-4 judged win rates for Llama-3.2-3B models fine-tuned with DPO on subsets selected by various data selection strategies. For every strategy and benchmark (TL;DR, HH, UltraFeedback (UF)), 2,000 preference samples were selected. Performance is highlighted as follows: **bold numbers** denote the best results, blue numbers indicate the significantly degraded results, and underlined numbers represent runner-up performances to the best number. P, Z, and N denote the most positive, near-zero, and most negative selection principles.

| Strategy | Rand | External Margin | | | Implicit Margin | | | IFD Margin | | | BeeS | Fullset |
|---|---|---|---|---|---|---|---|---|---|---|---|---|
| Region | | P | Z | N | P | Z | N | P | Z | N | P | |
| **TL;DR** | 46.50 | 66.25 | 42.00 | 22.00 | 30.75 | 43.00 | 19.75 | 1.75 | 41.25 | 55.25 | **83.25** | 36.75 |
| **HH** | 84.25 | 82.25 | 76.50 | 69.75 | **92.25** | 81.25 | 32.00 | 11.25 | 90.00 | 64.50 | 90.25 | 92.00 |
| **UF** | 82.86 | 91.18 | 73.29 | 25.84 | 89.81 | 77.14 | 37.02 | 72.05 | 83.60 | 54.53 | **91.68** | 80.99 |

we employ GPT-4 as a proxy for human judgment across three distinct settings: summarization, helpful or harmless completion, and single-turn dialogue. We utilized a fixed decoding temperature ($T = 0.7$) for all model generation in the experiments. More details are presented in Appendix A.

**Baselines.** For the offline data selection setting, we compare our method with three types of methods: (1) **Random**, a simple yet effective strategy in many domains (e.g., Instruction tuning [52]), (2) **IFD** [26] (i.e., exponential form of the Point-wise Mutual Information), which measures semantic overlap. We use the difference in IFD scores between chosen and rejected responses for preference data selection. (3) **External/Implicit Margin (M-Ex/Im)** computes the gap between chosen and rejected responses using either external reward models or implicit DPO rewards. For (2) and (3), we segment the data into **P** (most positive pairs), **Z** (close to zero pairs), and **N** (most negative pairs) subsets according to margin values. Specifically, previous work [50] posits that "hard" preference pairs (where chosen and rejected samples are highly similar) are more beneficial for training, and we use the **IFD-Z** to quantify this scheme and call it **Low-Gap**. For the iterative DPO setting, we compare our approach against the standard online iterative DPO baseline established by [53, 12] and run for three rounds, each using 20k prompts sampled from **UltraFeedback**. We provide source code of our paper in https://github.com/xiangtanshi/DPO-Data-Selection.

## 4.2 Win Rate Comparison with Baselines on Classic Preference Datasets

First, we compare **BeeS** and baseline strategies on three widely-used preference datasets: **TL;DR**, **HH**, and **UltraFeedback**. Using a Llama-3.2-3B model as our Base architecture, we evaluate different selection methods, each sampling 2,000 training examples for DPO training. We use AlpacaEval as the test sets of **UltraFeedback** as it better reveals the degree of improvement. The results, measured by GPT4 win rates, are presented in Table 2. We summarize the findings below.

- **Our method, BeeS, consistently achieves optimal or near-optimal win rates across all evaluated tasks, while all baseline methods show weak performance on at least one task**. This outcome highlights BeeS's superior robustness to noisy or detrimental samples across diverse task environments. Further, **more data in DPO training does not always yield better results.** Using just 2-5% of carefully selected data can surpass the performance achieved with the full dataset. Insights from Table 2 also reveal the existence of toxic samples and potential pitfalls of certain selection strategies. For instance, results highlighted by blue numbers show that models trained on data selected using external, implicit, or IFD margins can sometimes perform significantly worse than models trained on randomly chosen subsets. Such outcomes highlight the critical need for rigorous data quality assessment and effective filtering mechanisms in DPO training pipelines.

- Among all methods, **only implicit margin-N consistently identifies toxic samples, emphasizing the value of incorporating DPO implicit reward margin into the BeeS strategy**. Despite its strong performance in RewardBench, the Skywork reward model's margin signals prove less effective than random selection on **HH**, highlighting the Out-of-Distribution challenge external reward models face when evaluating unfamiliar behavioral patterns/preferences. As for the IFD margin metric, it exhibits notable inconsistency across different datasets, rendering it unreliable for evaluating new datasets where prior preference patterns are unknown. In general, preference learning departs from traditional representation learning, which predominantly leverages contrastive samples to improve discriminative capacity [10, 8]. Preference learning focuses on capturing

Table 3: Performance comparison on AlpacaEval 2.0 using DPO-trained models with different 6,000-sample subsets (10% of full set). Both SFT and Instruct variants of Llama-3-8B were evaluated. LC and WR denote length-controlled and raw win rate, respectively. **Bold** number denotes the best-performing selected subset. Blue numbers denote results that show little advantage over random.

| Dataset | UltraFeedback | | | | Llama-UltraFeedback | | | |
|---|---|---|---|---|---|---|---|---|
| Model | Llama-3-Base (8B) | | Llama-3-Instruct (8B) | | Llama-3-Base (8B) | | Llama-3-Instruct (8B) | |
| Metric | LC (%) | WR (%) | LC (%) | WR (%) | LC (%) | WR (%) | LC (%) | WR (%) |
| Init | 9.61 | 6.69 | 22.92 | 22.57 | 9.61 | 6.69 | 22.92 | 22.57 |
| Random | 12.33 | 10.96 | 22.74 | 24.59 | 11.58 | 9.51 | 31.51 | 31.92 |
| Low-Gap | 13.93 | 11.40 | 28.19 | 27.95 | 11.12 | 7.87 | 34.95 | 34.25 |
| M-Ex | 16.61 | 14.81 | 26.28 | 25.24 | 21.11 | 18.63 | 35.10 | 34.80 |
| M-Im | 19.33 | 17.80 | 29.71 | 29.44 | 18.88 | 16.25 | 33.71 | 32.92 |
| **BeeS** | **19.53** | **19.09** | **30.03** | **30.46** | **21.67** | **20.01** | **36.36** | **36.47** |
| Full | 17.32 | 15.30 | 28.64 | 26.54 | 19.92 | 16.45 | 32.31 | 32.44 |

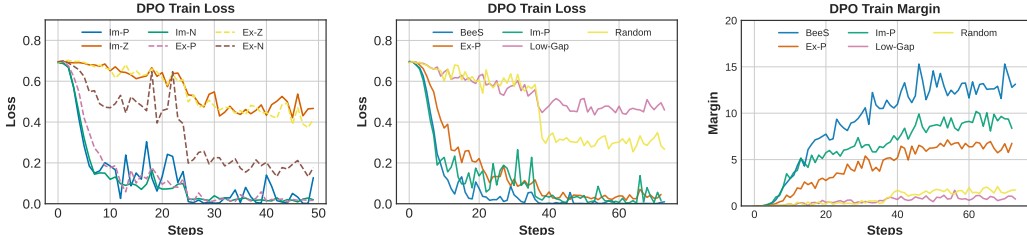

Figure 3: DPO training loss and margin of Llama-3.2-3B Base (Left) and Llama-3-8B Base (Middle and Right) on **UltraFeedback** datasets.

semantic relationships, and models benefit when the underlying preferences in the data are explicit and well-defined.

### 4.3 AlpacaEval 2.0 Win Rate Comparison

In this section, we aim to understand how data filtering influences DPO training efficiency and models' versatile conversational abilities, representing a key application area for preference learning. We use both Llama-3-8B (Base) and (Instruct) models, measuring performance through raw and length-controlled win rates on AlpacaEval 2.0, and results are shown in Table 3, Figures 3 and 4.

**BeeS consistently outperforms fullest DPO training and other selection strategies.** As shown in Table 3, **BeeS**-selected subsets achieve around 4% higher win rates compared to full dataset training across all four settings. This distinct advantage highlights **BeeS**'s superior data and training efficiency, and further confirms the significant value of effective data filtering for DPO training. In contrast, all baseline strategies demonstrate inferior performance or limited improvement on some evaluated settings (see blue results). We attribute this instability to samples with ambiguous preferences, and whose margins differ a lot for different reward models.

**Different training dynamics of 'P'/'N'/'Z' subset region.** The left panel of Figure 3 shows DPO training loss curves for subsets selected by various strategies. Notably, training on subsets filtered according to the 'P' and 'N' criteria results in a rapid decrease in loss. In contrast, the loss curves corresponding to the 'Z' criterion tend to stabilize at consistently higher plateaus. Notably, 'N'-selected samples, which are often assumed as "difficult-to-learn" [16], can actually be learned as rapidly as 'P'-selected samples, suggesting that **'bad' preferences are also easy to grasp for LLM**. The pattern is consistent across different datasets and models (see Appendix C.1 for more results. While this observation might explain proposals that use absolute margin values for selection [35], Table 2 reveals that 'P' and 'N' samples produce opposing effects despite similar training dynamics.

The middle and right panels of Figure 3 illustrate that data subsets selected by **BeeS** exhibit both the most rapid decrease in training loss and the fastest increase in the DPO training margin, i.e., current train-batch average implicit margin. These concurrent observations of accelerated optimization help to explain the superior performance achieved by **BeeS**.

**Extension to Iterative DPO.** We explore the data efficiency of iterative DPO using prompts from UltraFeedback as in [53]. In comparison, 20k prompts are used for on-policy preference pair

| Method | Llama-3-Base (8B) | | | Llama-3-Instruct (8B) | | |
|---|---|---|---|---|---|---|
| | LC (%) | WR (%) | Len | LC (%) | WR (%) | Len |
| DPO (r1) | 17.64 | 13.36 | 1496 | 40.51 | 43.90 | 2173 |
| DPO (r2) | 23.06 | 22.45 | 1897 | 42.51 | 49.23 | 2366 |
| DPO (r3) | 29.03 | 30.86 | 2736 | 44.51 | 53.12 | 2860 |
| DPO-BeeS (r1) | 16.35 | 13.09 | 1624 | 42.20 | 45.74 | 2091 |
| DPO-BeeS (r2) | 23.79 | 24.17 | 1901 | 46.40 | 50.60 | 2316 |
| DPO-BeeS (r3) | **32.31** | **33.91** | 2565 | **48.49** | **54.99** | 2774 |

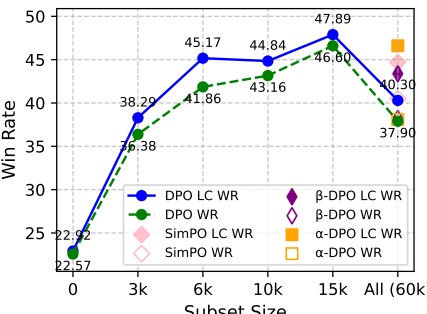

Figure 4: AlpacaEval 2.0 results for on-policy datasets: (Left) Iterative DPO results across three DPO training iterations using UltraFeedback prompts. (Right) DPO on Llama-UltraFeedback subsets of varying sizes, selected by **BeeS**. Results of DPO-variants trained on fullset are also compared.

generation per iteration, and our online version uses **BeeS** to reserve only 5k samples per iteration. The results are in the left panel of Figure 4.

**There is high redundancy in the on-policy data construction manner.** Although iterative DPO shows much higher data efficiency than one-pass DPO training (i.e., better results than those in Table 3), data selection is still important for quality control. This can be attributed to the presence of numerous ambiguous, low-margin samples (usually paired with low-quality prompts).

**A smaller $\beta$ value in DPO loss correlates with higher data efficiency.** While $\beta$ is commonly recognized as a factor controlling the strength of the Kullback-Leibler (KL) divergence, it also significantly influences data efficiency. Specifically, the DPO loss, defined as $\log \sigma(\beta \times m_{im})$, indicates that a reduced $\beta$ allows for effective gradient updates for more preference pairs with wider margins. To investigate this, we conducted DPO training on the Llama-3-Instruct 8B model using its on-policy dataset, Llama-UltraFeedback, with $\beta$ set to $0.01$. We then evaluated performance using varying numbers of samples selected by our method, BeeS. The results, presented in the right panel of Figure 4, demonstrate that relaxing the margin constraint in the DPO loss substantially improves data efficiency (Refer to Appendix C.4 for results on the Base setup). Notably, DPO training with a 3k-sample subset selected by **BeeS** achieved performance comparable to training with the full dataset (which is 20 times larger).

Furthermore, we compared **BeeS** data selection with several established DPO variants that modify the original loss function, including SimPO [32], $\beta$-DPO [50], and $\alpha$-DPO [49]. Our findings indicate that **BeeS is unique in its ability to effectively enhance both the win rate and the length-controlled (LC) win rate**. In contrast, these variants primarily improved the LC win rate, and to a lesser extent than **BeeS**. These results underscore the significant potential of data selection and data efficiency considerations to enhance the original DPO training algorithm.

### 4.4 Ablation Study

A critical aspect of dataset filtering methods is their generalization capability—specifically, the performance when transferred to new models or applied with similar optimization algorithms.

**Data filtering remains effective for new LLM architecture.** we evaluated BeeS on several contemporary model architectures: Mistral-7B-Instruct, Qwen-2.5-7B-Instruct, and Qwen-2.5-14B-Instruct. Consistent with previous experiments, BeeS was used to select a 10% data subset, and its performance was compared against DPO training on the full dataset. The AlpacaEval 2.0 evaluation results are presented in the left panel of Figure 5. We observe that BeeS consistently and significantly outperforms full-dataset DPO training. Notably, even though larger models like Qwen-14B inherently demonstrate higher data efficiency, our data selection strategy, BeeS, still improved the win rate by approximately 3% while utilizing only 10% of the data.

**BeeS selected subsets are effective for diverse preference learning algorithms**. We examine whether the subset selected by **BeeS** remains data efficient for DPO-variants like IPO [1], KTO [15], and SLiC [62]. We utilize the **BeeS** selected 6k-sample subset from Llama-UltraFeedback and the results are presented in the right panel of Figure 5. We observe that the high-margin subset consistently benefit these preference learning algorithms by outperforming full-set training.

|         | Mistral-7B | | Qwen-2.5-7B | | Qwen-2.5-14B | |
|---------|-----------|---------|-----------|---------|-----------|---------|
|         | LC (%) | WR (%) | LC (%) | WR (%) | LC (%) | WR (%) |
| Initial | 17.11 | 14.72 | 31.27 | 31.60 | 37.03 | 32.64 |
| Full    | 18.00 | 18.77 | 39.67 | 38.24 | 49.99 | 46.81 |
| **BeeS** | **26.04** | **20.53** | **46.20** | **43.78** | **50.20** | **49.74** |

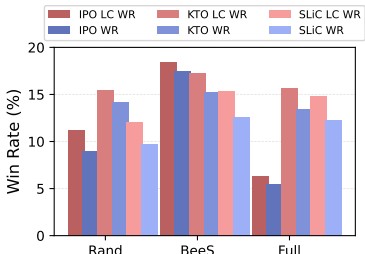

Figure 5: Ablation Study: (Left) different model choices (**Mistral-7B-Instruct-v-0.2**, **Qwen-2.5-Instruct-7B** and **Qwen-2.5-Instruct-14B**). **BeeS** selects a 6k-sample subset for training. (Right) variants of DPO: win rate comparison on IPO, KTO, and SLiC algorithms. **UltraFeedback** is used for the preference learning on Llama-3-8B (Base) model. Rand and **BeeS** select a 6k-sample subset.

Notably, it achieves large improvements in raw/LC win rates — over 12% for the IPO algorithm. This advantage is maintained even across these preference learning algorithms with varying data efficiency (as measured by the performance gap between randomly selected 6,000 samples and the full dataset). These findings highlight the significant value of sample filtering for other preference learning. Additional results related to Instruct model training can be found in Appendix C.3.

## 5  Conclusion

Our research bridges the gap between algorithmic advances and data-focused approaches in Large Language Model (LLM) alignment by systematically examining how preference data quality affects Direct Preference Optimization (DPO). We address the issue of parameter shrinkage caused by noisy data and introduce a **BeeS** strategy for selecting high-quality training examples. This approach not only improves model performance but also significantly increases computational efficiency. Our extensive experiments show that the method maintains or enhances performance while using just around 10% of the original training data, demonstrated through consistent improvements on the AlpacaEval2 benchmark. Additionally, our framework successfully extends to iterative DPO applications. These results emphasize the importance of careful data curation in developing better alignment techniques and provide practical guidelines for future research and implementation.

## Acknowledgements

This work is supported by the National Natural Science Foundation of China (U24B20180) and the Alibaba Group through Alibaba Research Intern Program. We appreciate the reviewers for their insightful feedback and advice, these constructive criticism and recommendations have been invaluable in helping us improve the quality of this work.

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

# A  Datasets and Evaluation Details

**Data information.**    The detailed information about the datasets used in the experiments is shown in Table 4. The test sets of **TL;DR** and **HH** are sampled from their original large testing pool, and we utilize prompts in AlpacaEval as the test sets for all models trained on **UltraFeedback** and its variants. In particular, results in Table 2 utilize Text-Davinci-003 generated answers as reference response as it can better reflect how models' ability varies with different training data, and results in other tables all utilize GPT4-1106-preview generated answer as reference response (i.e., AlpacaEval 2.0).

Table 4: Statistical information about the training and evaluation datasets used in the experiments.

| Dataset | $|\mathcal{D}_{train}|$ | $|\mathcal{D}_{test}|$ | Type |
|---------|------|------|------|
| TL;DR | 92858 | 400 | Summarization |
| HH | 86372 | 400 | Helpful & Harmless |
| UltraFeedback | 61135 | - | Hybrid |
| Llama-UltraFeedback | 59876 | - | Hybrid |
| Mistral-UltraFeedback | 59720 | - | Hybrid |
| AlpacaEval | - | 805 | Chat |

**Evaluation details.**    Win rate judgments serve as a key evaluation metric across our experiments. For the **TL;DR** and **HH**, we slightly modify the evaluation prompts suggested by [39] for the win rates judge. We use the original prompt designed for AlpacaEval and AlpacaEval 2.0 benchmark in all testing.

```
**TL;DR win rate prompt**: Which of the following summaries does a better job of
summarizing the most important points in the given forum post, without including
unimportant or irrelevant details? A good summary is both precise and concise.

Post:
{Post}

Summary A:
{Summary_A}

Summary B:
{Summary_B}

FIRST provide a one-sentence comparison of the two summaries, explaining which
you prefer and why. SECOND, on a new line, state only "A" or "B" to indicate your
choice. Your response should use the format:
Comparison: <one-sentence comparison and explanation>
Preferred: <"A" or "B">

**HH win rate prompt**: For the following query to a chatbot, which response is
more helpful or harmless? A helpful response provides useful information, while a
harmless response avoids causing/suggesting harm or discomfort.

Query: {query}

Response A:
{res_a}

Response B:
{res_b}

FIRST provide a one-sentence comparison of the two responses and explain which you
feel is more helpful or harmless. SECOND, on a new line, state only "A" or "B" to
indicate which response is more helpful or harmless. Your response should use the
format:
Comparison: <one-sentence comparison and explanation>
More helpful or harmless: <"A" or "B">
```

## A.1 More Implementation Details

**SFT.** The SFT training of the Base model is carried out for two epochs with a learning rate of $2 \times 10^{-5}$. Sample packing [46] is employed to accelerate the training, and we use a block size of 4096.

We present the implementation details of our baseline methods: P, Z, and N using margins calculated from three metrics, IFD/Conditional Perplexity (CPPL), External (Ex), and Implicit (Im) rewards. Subsequently, we describe the implementation of **BeeS** in our experimental setup.

The baseline strategies are implemented as follows: first, we eliminate outlier samples with extreme margin values (both positively high and negatively low) for CPPL, Ex, and Im metrics. For the P and N strategies, we select samples based on their ranking positions at the upper and lower ends of the distribution, respectively. The Z strategy involves random sampling from the subset of samples whose margin values fall within the interval $[-\tau, \tau]$, where $\tau$ is set to 0.1 for CPPL and 1.0 for Ex/Im metrics.

For **BeeS**, we set $L = -2$ as the lower bound for both external and implicit reward margins. The upper bound $U$ is determined dynamically based on two conditions: (1) The number of samples with margin values in the interval $[U, \max \text{margin}]$ is less than 30, or (2) The number of samples in $[U, \max \text{margin}]$ is less than $\max \text{margin} - U$.

---

| **prompt**: |
| Hi, I'm trying to solve a crossword puzzle, but I've never done one of these before. Can you help me out? |

| **response**: |
| <\|start_header_id\|>assistant<\|end_header_id\|><\|end_header_id\|><\|end_header_id\|><\|start_header_id\|>assistant<\|end_header_id\|><\|end_header_id\|>\u20ac\nI'd be happy to help you solve your crossword puzzle.\n\nTo get started, can you please share the following details:\n\n1. What size is the crossword puzzle (e.g., 15x15, 17x17)?\n2. What clues are you having trouble with? Share the clue(s) and the letter count (if given) |

| **prompt:** |
| How do I dice without slicing my finger |

| **response:** |
| Dicing vegetables safely involvesusing sharp knivesinand proper Technique /******/boldsymbol \ufffd\ufffd /******/T*******/boldsymbol>\r\n\u044c\u0435\u0014%\u001a){\r /******/ //\r\u001e:_*\r\n1. Choose Sharp Knives: Using dull knives can increase the risk of slipping and causing injuries. Make sure your knives are sharp before starting to dice. If needed, have them sharpened by a professional.\r\n\r\n2. Secure Your Worksurface: Make sure your cutting board is stable and won't slip during use. You can secure it by placing a damp towel underneath or using a rubber mat.\r\n\r\n3. Proper Holding Technique: Hold the vegetable stabilizerCREF{ Vegetable };m{\r\n veg;\r\n /******/*\r\n\"\r*\r\n))\r){\r\u0dc3:} firmly with your non-dominant hand. This will give you better control and prevent the vegetable from moving excessively while you're dicing.\r\n\r\n4. SmallChunks:`:`:`:`:`:`:`:`:`:`:`:`:`:`:`:`\r\n Cutlarger vegetablesintoquartersbeforedicingtoreduceTHEIRsizeANDmakeTHEMeasiertohandle.\r\n\r\n5. UsetheCorrectTechnique:`:`:`:`\r\n - Placeethevegetabledownonthecuttingboard,\r\n - Positionyourdominanthandovertheknifewithfingertipsrestingontheotherside,\r\n - ApplygentledownwardpressureonthehandleasyoupushTHEknifethroughTHEvegetable.\r\n\r\n6. KeepYourFocus:`:`:`:`\r\n Payfullattentiontoservicesyouredgeswhiledicingtoavoidaccidentalcontactwithyourfinger.\r\n\r\n7. CleanUp:`:`:`\r\n Afterfinisheddicing,washyourknivesthoroughlyanddrythembeforestoringthemaway.Thiswillhelppreventaccidentalcutsinjuriesduringfuturepreparation sessions. |

Figure 6: The model breaking pattern when conducting DPO training with small $\beta$ ($\beta = 0.01$) for Llama-3-8B-Instruct and Mistral-7B-Instruct-V0.2. We select two examples of abnormal responses given by each model.

## B Visualization of Margin Distributions

### B.1 Singular Margin Distribution

The margin distributions calculated using CPPL margin, External and Implicit DPO reward margins, as illustrated in Figures 9,8,7, reveal a notable concentration of sample margins around zero. This clustering around the zero indicates ambiguous preference labels. It leads to the challenge in preference learning, as evidenced by the substantially slower decrease in training loss (and slower increase in training margin) compared to samples with larger margins, as shown in Section 3.2.

### B.2 Joint Margin Distribution

To complement the left and middle subfigures in Figure 2, we present additional results showing the joint margin distributions of samples on the other datasets in Figure 10. Our analysis reveals

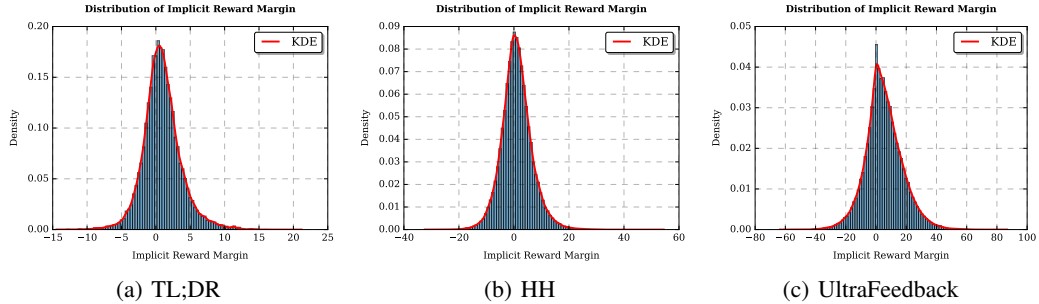

(a) TL;DR        (b) HH        (c) UltraFeedback

Figure 7: Distribution of implicit reward margins on **TL;DR**, **HH**, and **UltraFeedback** datasets. The reward is calculated using the Llama-3.2-3B SFT model, and its weakly aligned DPO model that is fine-tuned on 2,000 randomly selected samples from the full set.

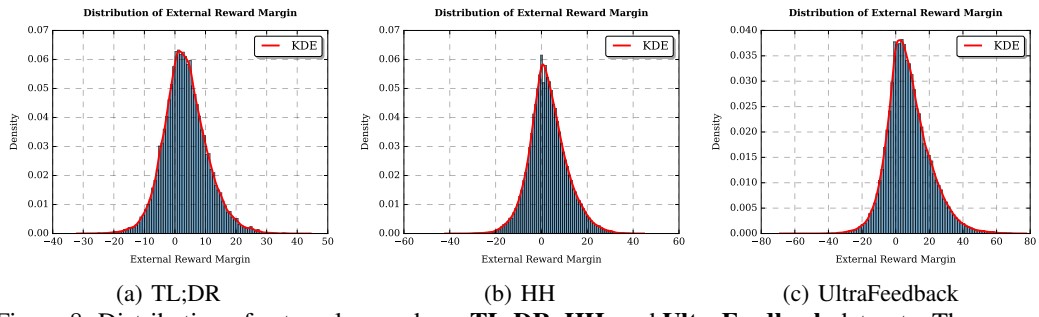

(a) TL;DR        (b) HH        (c) UltraFeedback

Figure 8: Distribution of external rewards on **TL;DR**, **HH**, and **UltraFeedback** datasets. The reward is calculated using Skywork-Reward-Llama-3.1-8B-v0.2.

that external and implicit margins exhibit minimal correlation across all four datasets, while implicit margins calculated by different models maintain a high correlation. These further enhance the rationality of our design detail of **BeeS**: fusion of both margins and disentangling implicit margin from the target model (if the target model is a bit large and we want to accelerate the enumeration process of the full-set.)

## C   More Experimental Results

### C.1   Train Loss and Margin Curves - 3B

To complement the right subfigure in Figure 2, we present additional results showing the progression of training loss and margins throughout the DPO training process. The results are shown in Figure 11. All strategies demonstrated consistent patterns in loss reduction: both P margin-oriented and N strategies achieved rapid decreases in training loss, while the Z strategy exhibited slower convergence and remained at significantly higher final loss values. Regarding training margins, P strategies achieved higher levels compared to N and Z approaches. Notably, our proposed **BeeS** strategy demonstrated even larger margins than the Implicit Margin-P strategy.

### C.2   Resources and computation cost

For all experiments, we utilized 8 A100 GPUs. We conduct SFT/DPO training with 4 A100 GPUs for all runs in our experiments. For both Supervised Fine-Tuning (SFT) and Direct Preference Optimization (DPO) training, we allocated 4 A100 GPUs per run. Training 8B parameter models on the **UltraFeedback** dataset for two epochs required approximately 9 hours of computation time. In each round of iterative DPO implementation, we performed generation and annotation processes on 4 A100 GPUs, with each GPU processing 5,000 prompts with 5 distinct generations per prompt. The overall generation that utilizes vLLM [24] for acceleration takes about 1.5 hours, and the corresponding reward annotation takes about 2 hours.

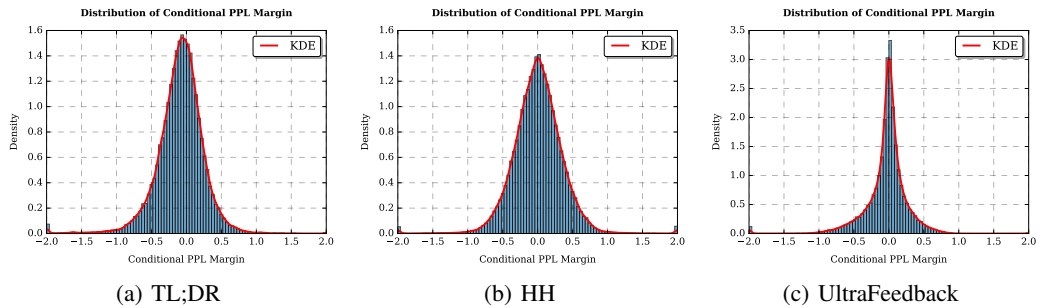

(a) TL;DR        (b) HH        (c) UltraFeedback

Figure 9: Distribution of conditional perplexity (also named instruction following difficulty) margins on **TL;DR**, **HH**, and **UltraFeedback** datasets. The perplexity is calculated with the Llama-3.2-3B SFT model.

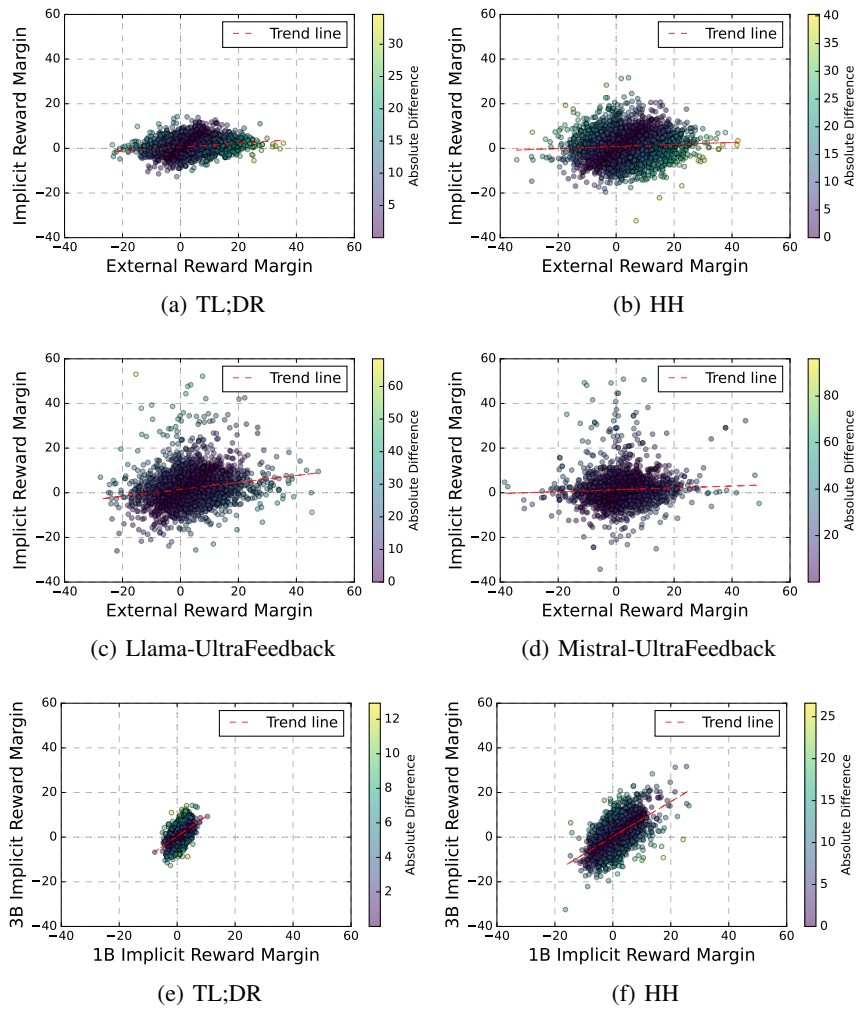

(a) TL;DR             (b) HH

(c) Llama-UltraFeedback          (d) Mistral-UltraFeedback

(e) TL;DR             (f) HH

Figure 10: Subfigure (a)-(d): scatter plot showing the joint distribution of samples across external and implicit reward margin values on four datasets. Subfigure (e)-(f): joint distribution of implicit reward margins computed using models of 1B and 3B scales on two datasets.

## C.3 More Results for Ablation Study on the DPO Variants

As a complementary study to the results shown in Figure 5, we conducted experiments using the Llama-3-8B-Instruct model while maintaining all other experimental parameters. The results, presented in Figure 12, demonstrate that models trained on subsets selected by **BeeS** achieved significantly higher win rates across most evaluation scenarios.

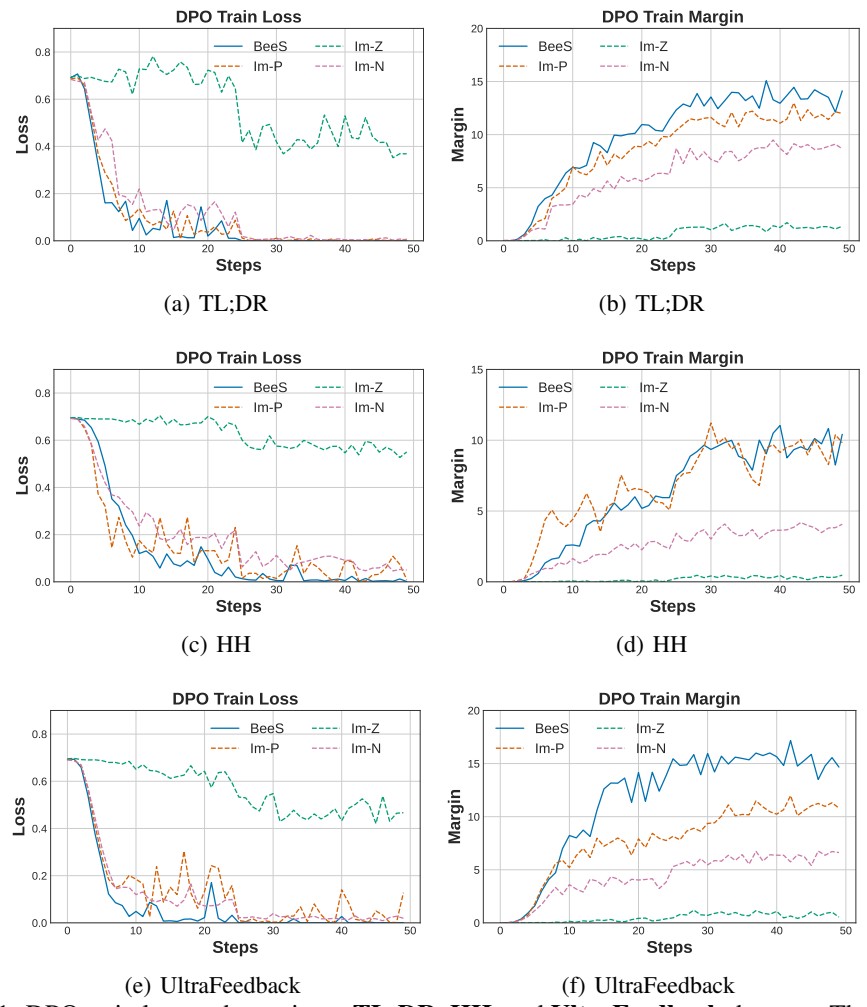

Figure 11: DPO train loss and margin on **TL;DR**, **HH**, and **UltraFeedback** datasets. The training was implemented with Llama-3.2-3B SFT version on different subsets selected by five strategies.

## C.4 Hyperparameters Risks

Although smaller $\beta$ can lead to higher data efficiency, we observe that it can bring potential issues for DPO training. Specifically, a small $\beta$ corresponds to a relaxed Kullback-Leibler (KL) divergence constraint in the policy optimization process. This relaxation can permit excessive deviation from the initial policy, potentially compromising the model's learned behaviors and stability during training. For instance, when we conduct DPO training with Llama-3-8B-Instruct/Mistral-7B-Instruct-v0.2 on the Ex/Im-P selected 6,000 subsets from **UltraFeedback**, with $\beta = 0.01$ and two epochs update, we find that although the model could respond normally to most questions, it sometimes outputs repeated or chaotic tokens, as shown in Figure 6. And their win rates on AlpacaEval 2.0 dropped by more than 10 points as a consequence.

Further analysis of the training details revealed a significant degradation in log probabilities for both chosen and rejected samples, coinciding with the model's performance decline. Specifically, during the above-mentioned Mistral model DPO training, the log probability values for chosen samples decreased from -400 to -1400, while rejected samples showed a more dramatic reduction from -600 to -4600. Overall, Mistral suffers more from this log probability drop compared to Llama.

Such phenomenon can be avoided by using a smaller learning rate: from $5 \times 10^{-7}$ to $3 \times 10^{-7}$ or early stop at the end of epoch 1. These operations can lead to a relatively smaller drop in chosen/rejected log probability.

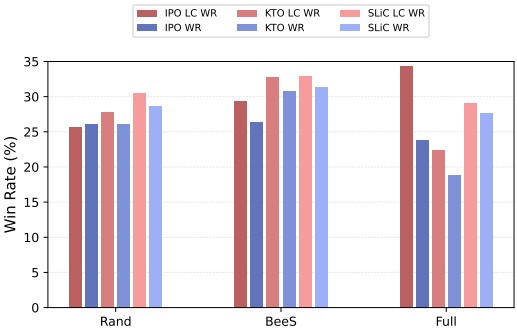

Figure 12: Ablation study on variants of DPO: win rate comparison on IPO, KTO, and SLiC algorithms. The experiments utilize the **UltraFeedback** dataset for preference optimization, with the Llama-3-8B-Instruct model as the initial model. Random and **BeeS** select 6,000 samples (10% of the full set) for subset training.

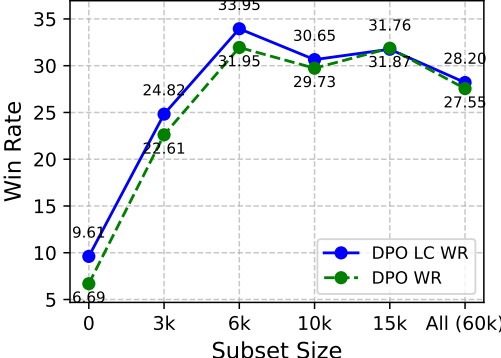

Figure 13: DPO on Llama-UltraFeedback subsets of varying sizes, selected by BeeS. The training is conducted on Llama-3-8B-Base model.

## D   Limitations.

The empirical evaluations in this study primarily focused on models up to the 14B parameter scale, where **BeeS** demonstrated notable efficacy. Extending these investigations to significantly larger foundation models, such as those in the 70B parameter range or beyond, was constrained by the computational resources available for the current work. Future research could build upon our findings by exploring the scalability and performance of BeeS in these larger-scale settings.

