# OpenReview forum: "Less is More: Improving LLM Alignment via Preference Data Selection"
_NeurIPS.cc/2025/Conference — NeurIPS 2025 spotlight_

### Official Review · Reviewer_8aHM · 2025-06-26

**Clarity:** 3
**Significance:** 2
**Originality:** 2
**Rating:** 4
**Confidence:** 3

**Summary:**

This paper claims that the reward margin is important for reward learning. They demonstrate the margin influence internally and externally. They introduce a method named BeeS to broaden the overall reward margin. The experiment shows the superiority of their method.

**Questions:**

Refer to above.

**Ethical Concerns:**

["NO or VERY MINOR ethics concerns only"]

**Final Justification:**

I lean to weakly acceptable.

**Limitations:**

Refer to above.

**Quality:**

2

**Strengths And Weaknesses:**

Strengths:

1. The idea and motivation are reasonable.

2. The method provides a theoretical explanation that appears reasonable.

3. The experimental results are strong.

Weaknesses:

1. In principle, pursuing a larger margin may improve reward learning, but when does a larger margin lead to better or worse outcomes? A theoretical and practical explanation is needed.

2. In Figure 1, how is the External Reward Margin calculated? You state that "the correlation between implicit and external reward margins is notably weak." Could this be due to inconsistent data distribution? Does this imply that the implicit reward results are consistent? Does the low correlation between the outputs of the two reward models in the right subplot of Figure 1 further support this point?

3. You mention using the BeeS method with three steps for low-margin cases. Including a small diagram illustrating the general process would enhance clarity.

4. In the Bayesian aggregation section, what are the advantages of this approach compared to traditional methods like averaging, finding maximum/minimum values, or Pareto optimization? Fundamentally, this is a reward margin fusion problem.

5. For well-trained models like Qwen3, compared to less well-trained models like Qwen2, does the effectiveness of your method diminish? When does your method perform best, and when is its effect less pronounced?

---

> ### Author Rebuttal · Authors · 2025-07-30
>
> We thank the reviewer for the encouraging feedback and insightful questions.
>
> **Weakness 1:** In principle, pursuing a larger margin may improve reward learning, but when does a larger margin lead to better or worse outcomes? A theoretical and practical explanation is needed.
>
> **Answer:** Thank you for your question. As discussed in Section 3.1, our theoretical analysis demonstrates that exogenous label errors cause parameter shrinkage toward zero, while larger margins provide a counterbalancing effect through parameter inflation, enabling correct parameter learning. Generally, larger margin preference pairs improve DPO training and accelerate convergence. However, excessive optimization can cause overfitting or model breakdown when margins become too large (e.g., training for many epochs), which is discussed in Appendix C.4.
>
>
>
> **Weakness 2:** In Figure 1, how is the External Reward Margin calculated? You state that "the correlation between implicit and external reward margins is notably weak." Could this be due to inconsistent data distribution? Does this imply that the implicit reward results are consistent? Does the low correlation between the outputs of the two reward models in the right subplot of Figure 1 further support this point?
>
> **Answer:** In Figure 1 (left panel), we use the skywork-reward-llama-3.1-8B-v2 to calculate the external reward margin as $r(x,y_w)-r(x,y_l)$. For the rest questions concerning consistency or inconsistency between reward models, we agree with you that it stems from similar or different training data distribution. Moreover, we should note that inconsistency primarily occurs on samples with ambiguous preferences.
>
>
>
> **Weakness 3:** You mention using the BeeS method with three steps for low-margin cases. Including a small diagram illustrating the general process would enhance clarity.
>
> **Answer:** Thank you for this helpful suggestion. We have made a flowchart to better illustrate algorithm details, including the three steps of margin calculation and margin aggregation with concrete examples showing how low-margin samples are deprioritized during selection. This flowchart will be included in the revised manuscript.
>
>
> **Weakness 4:** In the Bayesian aggregation section, what are the advantages of this approach compared to traditional methods like averaging, finding maximum/minimum values, or Pareto optimization? Fundamentally, this is a reward margin fusion problem.
>
> **Answer:** The advantage is that Bayesian aggregation appropriately penalizes samples with at least one source of low margin while incorporating margin information from all sources. For example, average and max strategies may incorrectly rank samples with margins like (0.2, 0.6) versus (0.35, 0.45), while the min strategy cannot distinguish between (0.6, 0.8) and (0.6, 0.9). As for Pareto optimization, it is difficult to define a proper Pareto solution in this context. To further support our claim, we conducted an ablation study on different margin fusion strategies. Results below show that BeeS outperforms other heuristic strategies.
>
>
> | fusion strategy | Llama-3-8B-Instruct |  | Mistral-7B-Instruct |  |
> | -------- | -------- | -------- |-------- | -------- |
> |      |   LC WR   |  WR   |  LC WR  |  WR   |
> |   average   |  35.97    |  35.79   |   23.67 | 18.21 |
> |   max   |   33.99  | 33.04  | 24.56 | 18.09 |
> |   min   | 35.49 | 35.10 |25.19 | 19.00|
> |   BeeS   | 36.36 | 36.47 |26.04 |20.53|
>
> **Weakness 5:** For well-trained models like Qwen3, compared to less well-trained models like Qwen2, does the effectiveness of your method diminish? When does your method perform best, and when is its effect less pronounced?
>
> **Answer:** The Qwen-3 series are thinking models that have undergo lots of post-training for various tasks including dialogue. We tested DPO training on Llama-UltraFeedback using Qwen-3-8B (disable its thinking mode), with AlpacaEval2.0 results (i.e., LC win rate and win rate) shown below. DPO struggles to effectively enhance strong thinking models. We believe this is reasonable since these models have typically undergone internal post-training (including DPO, RLHF, etc.), while DPO and data selection perform best for improving base model performance or on new tasks that have not been post-trained.
>
>
>
> | Original | Fullset | BeeS top 6k |
> | -------- | -------- | -------- |
> |  (58.8, 63)   |  (54.33, 67.65)    |  (57.56, 68.12)   |

---

> > ### Comment · Reviewer_8aHM · 2025-08-06
> >
> > Thank you for your response—it has addressed some of my concerns. However, I will maintain my original score.

---

### Official Review · Reviewer_NjgC · 2025-06-29

**Clarity:** 3
**Significance:** 3
**Originality:** 3
**Rating:** 5
**Confidence:** 4

**Summary:**

This paper proposes a novel data-centric approach to improve the alignment of Large Language Models (LLMs) through Direct Preference Optimization (DPO) by emphasizing preference data selection instead of modifying the learning objective. The authors introduce a method called BeeS (Bayesian Aggregation for Preference Data Selection), which selects training samples by aggregating multiple reward signals and prioritizing those with consistent, high preference margins. The approach is theoretically motivated and empirically validated across various LLMs and datasets.

**Questions:**

1) Can you add SimPO to Figure 4 (right)?

2) Have you tried comparing with a strategy similar to that used for argilla/dpo-mix-7k? That is only selecting preference pairs where the chosen response has a high score.

3) A known issue of DPO is that it pushes probability mass away from the preference dataset and to unseen responses [1]. Is this behavior still present in your methodology?

[1] Shusheng Xu, Wei Fu, Jiaxuan Gao, Wenjie Ye, Weilin Liu, Zhiyu Mei, Guangju Wang, Chao Yu, and Yi Wu. Is dpo superior to ppo for llm alignment? a comprehensive study, 2024

**Ethical Concerns:**

["NO or VERY MINOR ethics concerns only"]

**Limitations:**

Yes

**Paper Formatting Concerns:**

They forgot to delete the first page of the checklist

**Quality:**

3

**Strengths And Weaknesses:**

Strengths:
- The paper tackles an underexplored aspect of preference-based alignment—data quality and selection—shifting the focus from algorithmic to dataset-level optimization.
- The authors provide mathematical analysis demonstrating how reward model noise leads to parameter shrinkage and how large-margin data selection counteracts this.
- The method is relatively simple.
- The authors provide extensive testing of their proposed method.

Weaknesses:
- The methodology relies on a full training run of DPO.
- No confidence intervals in the results. Given that the methodology relies on two different DPO runs (one on the full dataset and one on the selected sub-dataset) I expect a lot of variability in the results. It would be good to understand how consistent are the results of this methodology.

---

> ### Author Rebuttal · Authors · 2025-07-30
>
> We thank the reviewer for the encouraging feedback and constructive suggestions.
>
> **Weakness 1:** The methodology relies on a full training run of DPO.
>
> **Answer:** Our method does not require full dataset DPO training. During the *in-distribution pre-DPO training* phase, we randomly select a small seed dataset of only 2,000 samples for DPO training—significantly smaller than the full datasets (60k, 80k, and 90k samples for UltraFeedback, HH, and TL;DR). We conduct full dataset DPO training solely as a baseline comparison to demonstrate our method's effectiveness and efficiency.
>
>
> **Weakness 2:** No confidence intervals in the results. Given that the methodology relies on two different DPO runs (one on the full dataset and one on the selected sub-dataset) I expect a lot of variability in the results. It would be good to understand how consistent are the results of this methodology.
>
> **Answer:** Thank you for your suggestion. We agree that multiple uncertain factors exist in the subset selection process, but our Bayesian aggregation of different reward margins ensures relatively stable final subsets and DPO training outcomes. We provide ablation results using varying numbers of reward sources (K=1, 2, 3) with Llama3-8B instruct and base models on the Llama-UltraFeedback dataset. The results demonstrate stability when K≥2, confirming the robustness of our approach.
>
> | K | Ins-8B |  | Base-8B |  |
> | -------- | -------- | -------- |-------- | -------- |
> |      | LC WR     | WR     | LC WR     | WR     |
> | $r_{im,3b}$    |   33.71   |   32.92   | 18.88 | 16.25 |
> | $r_{im,3b},r_{ex}$     | 36.36     | 36.47     | 21.67 | 20.01 |
> | $r_{im,3b}, r_{ex}, r_{im,8b}$     | 36.53     | 37.47     | 21.83 | 20.37 |
>
> Regarding confidence intervals, AlpacaEval2.0 evaluation using GPT-4 API is stable (the standard error of Win Rate mostly fall within $(1.0,1.5)$) but expensive (more than $10 per evaluation). Due to these reasons, most previous work has not conducted multiple evaluations for confidence interval calculations.
>
> **Question 1:** Can you add SimPO to Figure 4 (right)?
>
> **Answer:** Thanks for your suggestion. The SimPO results, (LC Win Rate, Win Rate), for Figure 4 (right) setup is as follows, and we will add it to the revised manuscript.
>
>
> | Fullset | Random 6k | BeeS 6k |
> | -------- | -------- | -------- |
> | (20, 18.72)     | (12.73, 11.3)     | (20.53, 20.12)   |
>
>
>
> **Question 2:** Have you tried comparing with a strategy similar to that used for argilla/dpo-mix-7k? That is only selecting preference pairs where the chosen response has a high score.
>
> **Answer:** Thank you for your suggestion. We found that in preference pairs where the chosen response has top scores, the average margins are more than one times smaller than those of top-margin preference pairs. For instance, the average margins of selected 6000 samples by BeeS and the strategy is 18.46 and 8.89 for Llama-UltraFeedback dataset. Additionally, results using this approach are 2-5 points lower than the BeeS strategy.
>
>
> **Question 3:** A known issue of DPO is that it pushes probability mass away from the preference dataset and to unseen responses. Is this behavior still present in your methodology?
>
> **Answer:** Thank you for raising this interesting question. After read this great article, we are aware that DPO tends to generate biased policies favoring out-of-distribution responses due to its off-policy training nature, and can be alleviated by iterative DPO. Our method likely retains this behavior since we focus on data selection rather than addressing the fundamental on-policy versus off-policy training difference. The phenomenon may be reduced but not eliminated in our approach, and we will add this discussion to the Related Work section in the revised manuscript.

---

> > ### Comment · Reviewer_NjgC · 2025-08-05
> >
> > Thank you for the clarifications.

---

### Official Review · Reviewer_Z5aN · 2025-07-02

**Clarity:** 2
**Significance:** 3
**Originality:** 2
**Rating:** 5
**Confidence:** 3

**Summary:**

The paper presents a data selection method for DPO, which uses a Bayesian Aggregation approach to select data with maximum margins. The authors first use theoretical analysis to motivate and justify why exogenous error can lead to parameter shrinkage and hurt learning, then propose a margin-based data selection method to compensate parameter shrinkage. With some analysis and comparison of reward margin from different sources, the authors conclude to use an aggregation of multi-source margins to perform data selection. The method is applied to both one-pass DPO and iterative DPO. Experiments with multiple LLM models and datasets demonstrate that the proposed data selection outperforms existing data selection baselines, and also outperforms training with the full dataset.

**Questions:**

- Can the method be applied to PPO and other online RL algorithms?
- When using reinforcement learning with verifiable reward, where the reward is relatively accurate, will it still be beneficial to do data selection? How does it fit into the current framework?

**Ethical Concerns:**

["NO or VERY MINOR ethics concerns only"]

**Final Justification:**

Thank the authors for detailed response. It sounds good to me. Given the interesting analysis and good results, I decide to raise my score to accept. Look forward to seeing the final version with polished writing as promised.

**Limitations:**

Appendix D discusses some limitations about scaling up.

**Quality:**

3

**Strengths And Weaknesses:**

### Strengths
- The paper is in general well-presented. The idea is clearly motivated and makes intuitive sense.
- The proposed method is solid with theoretical interpretation. The numerical analysis in Figure 1 is also interesting.
- The experiment results look promising.

### Weakness
- The idea of selecting preference data with large margin doesn't sound very new to me. But I find the analysis in this paper helpful and grounded.
- It would be better if the method can be extended to broader scenarios, not only DPO. For example, does it apply to online RL algorithms like PPO?
- The algorithm presented in Section 3.2 is a little complicated and expensive with multiple steps and calculations. It is also not very clear what are the K reward sources used and how to select them in practice. It would be nice if the authors can describe it in more details and do some ablation study in experiments.
- The notations can be made more clear and simpler in section 3.1.

---

> ### Author Rebuttal · Authors · 2025-07-30
>
> We thank the reviewer for the encouraging feedback and valuable comments.
>
> **Weakness 1:** The idea of selecting preference data with large margin doesn't sound very new to me. But I find the analysis in this paper helpful and grounded.
>
> **Answer:** Thank you for acknowledging the value and rigor in our analysis. While margin-based filtering has been mentioned in concurrent work, these studies do not systematically analyze how different aspects of margin (e.g, sign, noise, and aggregation manner) impact data quality and DPO training. Our paper is the first to provide clear criteria for selecting informative samples and filtering out toxic ones, addressing this gap in the current literature.
>
>
> **Weakness 2 and Question 1:** It would be better if the method can be extended to broader scenarios, not only DPO. For example, does it apply to online RL algorithms like PPO?
>
> **Answer:** Thank you for your question. We answer your question from two perspectives: (1) Data selection remains important for efficient PPO training, and margin-based guidance can extend to RL scenarios. To illustrate, we compared two prompt sets: BeeS-selected 2k samples from UltraFeedback versus randomly selected 2k samples, performing RLHF on our SFT version of Llama-3.2-3B (using PairRM as the reward model). The AlpacaEval1.0 win rates were 88.84% for BeeS versus 78.12% for random (initial win rate is 61.49%). (2) While PPO on carefully selected prompt subsets converges faster, validating whether it consistently outperforms full-set PPO training across larger models and better reward models requires more rigorous experiments. We leave this for future exploration.
>
>
> **Weakness 3:** The algorithm presented in Section 3.2 is a little complicated and expensive with multiple steps and calculations. It is also not very clear what are the K reward sources used and how to select them in practice. It would be nice if the authors can describe it in more detail and do some ablation study in experiments.
>
> **Answer:** Thanks for your suggestion. We infer that you think the algorithm details are computationally expensive and the writing should be easier to follow. First, we have made a flowchart to better illustrate algorithm details (i.e., three steps of margin calculation and margin aggregation with concrete examples showing how low-margin samples are deprioritized during selection) and will add it to the revised manuscript. Second, we claim that the compute costs are acceptable for two key reasons: pre-DPO training requires only a small model trained on a randomly selected subset (size of 2k samples); margin calculations represent a one-time computational investment, and the selected subset can be used for training any subsequent model.
>
> In our experiments, we employed two reward sources following a straightforward selection principle: including at least one implicit and one external reward model. We conducted an ablation study on K using the Llama3-8B instruct and base models with the Llama-UltraFeedback dataset. We evaluated results when using one to three reward sources, and we observed that integrating additional reward sources yields diminishing returns when $K\geq 2$, hence we recommend K=2.
>
> | K | Ins-8B |  | Base-8B |  |
> | -------- | -------- | -------- |-------- | -------- |
> |      | LC WR     | WR     | LC WR     | WR     |
> | $r_{im,3b}$    |   33.71   |   32.92   | 18.88 | 16.25 |
> | $r_{im,3b},r_{ex}$     | 36.36     | 36.47     | 21.67 | 20.01 |
> | $r_{im,3b}, r_{ex}, r_{im,8b}$     | 36.53     | 37.47     | 21.83 | 20.37 |
>
> **Weakness 4:** The notations can be made more clear and simpler in section 3.1.
>
> **Answer:** Thank you for your suggestion. We use the concise table below to explain the meaning of the notation.
>
> | r | x | $y_w$ / $y_l$ | $\phi$ / $\Delta\phi$| $\zeta$|$\omega$|
> | -------- | -------- | -------- |-------- |-------- |-------- |
> | Reward    | 	Input prompt    | Preferred / dispreferred response  |  	Feature function in the reward model / relative feature function | 	Exogenous error in preference| 	Learnable parameters|
>
> We will include this table in the revised version to help readers better understand the notations and ideas. Thank you!
>
>
> **Question 2:** When using reinforcement learning with verifiable reward, where the reward is relatively accurate, will it still be beneficial to do data selection? How does it fit into the current framework?
>
> **Answer:** Thank you for raising this important point. We argue that data selection remains beneficial in RLVR contexts for two key reasons: (1) Removing overly easy or difficult prompts (analogous to near-zero margin cases in DPO) significantly accelerates RLVR convergence, as supported by dynamic sampling results in DAPO [1]. (2) Targeting prompts where the model's rollout accuracy approaches 50% ensures exposure to cases with the largest reward contrast. This is similar in spirit to prioritizing large-margin examples in DPO: it creates a clear distinction between frequently generated incorrect chains-of-thought and correct ones. As above, data selection can be naturally fit into current RLVR: keeping those learnable prompts that LLM rollout accuracy is near 50%.
>
> **Limitation** Appendix D discusses some limitations about scaling up.
>
> **Answer:** Regarding the applicability of our method to larger models in DPO training, recent studies [2] have demonstrated that strategic data selection can significantly improve dialogue performance in models as large as the 70B Llama model. We are confident that our method holds promise for DPO training at such scale, and plan to investigate this further in future work when the resource constraint problem is solved.
>
> [1] Yu, Qiying, et al. Dapo: An open-source llm reinforcement learning system at scale. (2025).
>
> [2] P Yu, W Yuan, et al. Rip: Better models by survival of the fittest prompts. (2025).

---

### Official Review · Reviewer_18xZ · 2025-07-03

**Clarity:** 3
**Significance:** 3
**Originality:** 3
**Rating:** 5
**Confidence:** 2

**Summary:**

This paper presents a novel preference data selection method designed to enhance DPO training effectiveness and efficiency. Specifically, the authors address noise reduction through a margin-maximization principle and propose a Bayesian Aggregation method to unify multiple margin sources (both external and implicit). Using only 10% of the original data, their approach achieves a 3-8% performance improvement.

**Questions:**

See above

**Ethical Concerns:**

["NO or VERY MINOR ethics concerns only"]

**Final Justification:**

This paper is solid with novel method and superior performance. I appreciate the authors' effort to address my concerns.

**Limitations:**

See above

**Quality:**

3

**Strengths And Weaknesses:**

**Strengths**

1. The paper introduces an innovative preference data selection method, the Bayesian Aggregation for Preference data Selection (BeeS) strategy, which incorporates signals from both external rewards and DPO implicit rewards. It deprioritizes preference pairs exhibiting low reward margins from any single reward source to mitigate potential noise.
2. The authors provide a detailed analysis of Parameter Shrinkage and Inflation Analysis, highlighting how inaccuracies in the reward model can cause LLM parameters to shrink toward zero.
3. The paper includes comprehensive comparative experiments and analyses that demonstrate the method's effectiveness. Using only 10% of the data, the approach achieves a 3-8% performance improvement, with significant performance gains even in online DPO scenarios.


**Weaknesses**

1. The proposed method is somewhat limited in scope, as the analysis and experiments focus exclusively on the DPO method. It remains unclear whether the approach can be extended to other, potentially superior RL methods such as PPO, GRPO, etc.
2. Minor concern: The authors do not address whether their method is applicable to thinking models or whether it would be equally effective for larger models.

---

> ### Author Rebuttal · Authors · 2025-07-30
>
> We thank the reviewer for the encouraging feedback and insightful comments.
>
> **Weakness 1:** The proposed method is somewhat limited in scope, as the analysis and experiments focus exclusively on the DPO method. It remains unclear whether the approach can be extended to other, potentially superior RL methods such as PPO, GRPO, etc.
>
> **Answer:** Thank you for your question. We answer your question from two perspectives: (1) Data selection remains important for efficient PPO training, and margin-based guidance can extend to RL scenarios. To illustrate, we compared two prompt sets: BeeS-selected 2k samples from UltraFeedback versus randomly selected 2k samples, performing RLHF on our SFT version of Llama-3.2-3B (using PairRM as the reward model). The AlpacaEval1.0 win rates were 88.84% for BeeS versus 78.12% for random selection (initial win rate is 61.49%). (2) While PPO on carefully selected prompt subsets converges faster, validating whether it consistently outperforms full-set PPO training across larger models and better reward models requires more rigorous experiments. We leave this for future exploration.
>
> **Weakness 2:** Minor concern: The authors do not address whether their method is applicable to thinking models or whether it would be equally effective for larger models.
>
> **Answer:** We appreciate the reviewer raising this point. At present, Reinforcement Learning approaches, such as RLVR and RLHF, remain the primary methods for enhancing thinking models. Our proposed method, however, is specifically tailored for DPO training (with potential extensions to PPO as suggested in the above answer), which is not typically used for thinking models. Regarding the applicability of our method to larger models in DPO training, recent studies [1] have demonstrated that strategic data selection can significantly improve dialogue performance in models as large as the 70B Llama model. We are confident that our method holds promise for DPO training at such scale, and plan to investigate this further in future work when resource constraint problem is solved.
>
> [1] P Yu, W Yuan, et al. Rip: Better models by survival of the fittest prompts. (2025).

---

### Decision · Program_Chairs · 2025-09-17

**Decision:**

Accept (spotlight)

**Comment:**

(a) Summary of Scientific Claims and Findings:
The paper introduces a novel data selection method, BeeS, for improving Direct Preference Optimization (DPO) training in large language models. It proposes a Bayesian aggregation approach to select training samples with high reward margins, aiming to reduce noise and improve alignment with human preferences. Experiments demonstrate significant performance gains using only a fraction of the original dataset, highlighting the efficiency and effectiveness of the proposed method.

(b) Strengths of the Paper:
Novel Data Selection Method: The BeeS method is innovative and provides a principled way to select high-quality training samples.
Theoretical Analysis: The paper offers a detailed analysis of parameter shrinkage and inflation, providing a strong theoretical foundation.
Empirical Validation: Comprehensive experiments across multiple models and datasets demonstrate consistent improvements over baseline methods.

(c) Weaknesses of the Paper:
Limited Scope: The method is currently limited to DPO, and its applicability to other RL methods like PPO is not fully explored.
Complexity of Algorithm: The multi-step Bayesian aggregation process may be computationally expensive and complex to implement.
Lack of Confidence Intervals: The results lack confidence intervals, making it difficult to assess the variability and reliability of the findings.

(d) Reasons for Decision:
The paper presents a novel and effective approach to data selection in DPO, supported by both theoretical analysis and empirical results. The significant performance improvements and data efficiency demonstrated in the experiments are compelling. However, the limited scope and complexity of the method suggest that further exploration and simplification are needed for broader adoption.

(e) Discussion and Rebuttal:
During the rebuttal period, the authors addressed several concerns raised by the reviewers. They provided additional experiments and analyses to demonstrate the potential applicability of their method to other RL algorithms like PPO. They also clarified the computational costs and provided a flowchart to enhance the clarity of the algorithm. These responses were generally satisfactory, reinforcing the paper's strengths and addressing some of the weaknesses.